# Validation of IL-7R as an Immunological Biomarker for Human Pancreatic Ductal Adenocarcinoma

**DOI:** 10.3390/cancers14030853

**Published:** 2022-02-08

**Authors:** Sung-Ill Jang, Jae-Hee Cho, So-Young Kim, In-Young Hong, Joon-Seong Park, Hye-Sun Lee, Goeun Park, Jong-Kyoung Kim, Hyung-Keun Lee, Dong-Ki Lee

**Affiliations:** 1Department of Internal Medicine, Gangnam Severance Hospital, Yonsei University College of Medicine, Seoul 06273, Korea; aerosij@yuhs.ac (S.-I.J.); jhcho9328@yuhs.ac (J.-H.C.); hdoll91@yuhs.ac (I.-Y.H.); 2Institute of Vision Research, Department of Ophthalmology, Yonsei University College of Medicine, Seoul 06273, Korea; sykim@acurasysbio.com; 3Department of Surgery, Gangnam Severance Hospital, Yonsei University College of Medicine, Seoul 06273, Korea; jspark330@yuhs.ac; 4Biostatistics Collaboration Unit, Yonsei University College of Medicine, Seoul 06273, Korea; hslee1@yuhs.ac (H.-S.L.); gogoeun@yuhs.ac (G.P.); 5Department of New Biology, Daegu Gyeongbuk Institute of Science & Technology, Daegu 42988, Korea; jkkim@digist.ac.kr

**Keywords:** pancreatic ductal adenocarcinoma, interleukin-7 receptor, cytokine, diagnosis

## Abstract

**Simple Summary:**

Despite being the fourth-leading cause of cancer-related deaths worldwide, pancreatic ductal adenocarcinoma (PDAC) lacks early diagnostic methods. We performed mRNA sequencing on peripheral blood mononuclear cells isolated from PDAC patients and identified IL-7R as a potential early diagnostic biomarker for PDAC. Furthermore, we found that IL-7R had improved diagnostic performance when combined with CA19-9. Our previous study’s results with 23 individuals were validated in a cohort of 522 patients. Our findings suggested that IL-7R in combination with CA19-9 could have important clinical implications that contribute to an earlier PDAC diagnosis and improved patient survival.

**Abstract:**

Pancreatic ductal adenocarcinoma (PDAC) is an aggressive cancer for which no early diagnostic method is available. The immune surveillance hypothesis suggests that the immune system plays crucial roles in tumor development and progression. We validated a PDAC-specific biomarker derived from peripheral blood mononuclear cells (PBMCs) to facilitate early PDAC diagnosis. mRNA levels of interleukin-7R (IL-7R), reportedly a potential immunological marker for PDAC, were measured in PBMCs isolated prospectively from healthy controls (n = 100) and patients with PDAC (n = 135), pancreatic cysts (n = 82), chronic pancreatitis (n = 42), acute pancreatitis (n = 47), and other malignancies (n = 116). The IL-7R level was significantly higher in patients with PDAC than in healthy controls, patients with benign pancreatic disease, and patients with other malignancies. As diagnostic parameters, the sensitivity, specificity, positive predictive value, negative predictive value, and accuracy for IL-7R were 58.5%, 92%, 90.8%, 62.2%, and 72.8%, respectively. The area under the receiver operating characteristic curve (AUROC) was 0.766. IL-7R levels did not differ between resectable and unresectable PDAC cases. The combined measurement of IL-7R and carbohydrate antigen 19-9 (CA19-9) significantly improved the diagnostic parameters and AUROC compared with the use of IL-7R or CA19-9 alone. IL-7R is significantly upregulated in PBMCs in patients with PDAC, and it may be a novel diagnostic marker for PDAC. The combined use of IL-7R and CA19-9 enhanced the diagnostic performance.

## 1. Introduction

Pancreatic ductal adenocarcinoma (PDAC) is the fourth leading cause of cancer-related mortality worldwide and accounts for 6% of annual cancer deaths. Although various treatment strategies for PDAC have been developed, the 5-year survival rate remains 3–5% [1,2,3,4]. Most patients with PDAC are asymptomatic until the disease progresses, and it metastasizes in approximately 50% of cases. Only 10–20% of patients with PDAC are candidates for surgical resection, which is the only treatment option that offers a potential cure [1,2].

To improve PDAC prognosis and treatment, early detection and surgical resection are crucial. However, the lack of diagnostic biomarkers for pancreatic cancer hampers the early diagnosis of this disease. Although carbohydrate antigen 19-9 (CA19-9) is a diagnostic marker for PDAC, [5] it has low diagnostic accuracy. CA19-9 assessment does not enable the discrimination of PDAC from other benign diseases, and false-positive results may be obtained in the presence of host inflammatory responses and obstructive jaundice. CA19-9 analysis also generates false-negative results in patients who are Lewis antigen-negative [5].

Crosstalk occurs between cancer and immune cells during cancer development and progression. [6] In addition, inflammatory and angiofibrotic cytokines are unregulated and critical for PDAC development and progression [7]. Thus, researchers have focused on the identification of PDAC-specific blood factors that could be used to detect tumors, disease progression, and relapse. [8] However, no biomarker other than CA19-9 is used clinically for PDAC detection or assessment of the progression of this disease.

Few studies have involved investigation of the use of blood cells from patients with PDAC to identify biomarkers. We used omics and bioinformatics techniques to identify blood factors with clinical implications for PDAC detection and interleukin-7R (IL-7R) was discovered as a PDAC-specific immunologic marker by total mRNA sequencing (mRNA-Seq) in a previous study [9]. We assessed the PDAC diagnostic efficacy of IL-7R in this study.

## 2. Materials and Methods

### 2.1. Study Design

This prospective case-control study was performed with patients with PDAC, healthy controls, patients with benign pancreatic disease, and patients with other malignancies. The study protocol was approved by the Institutional Review Boards of Gangnam Severance Hospital (no. 3-2018-0293) and was registered at the clinical research information service (https://cris.nih.go.kr/cris/en/; KCT0004614, access on 8 January 2020). The study was conducted according to the Helsinki declaration (2008, amended version), and written informed consent was obtained from each patient preoperatively and from all control participants.

### 2.2. Patients

We performed a pilot study with healthy controls and patients with PDAC to determine levels of blood cell markers [9]. We validated the expression levels of the blood cell markers in a large population. We collected peripheral blood samples from patients diagnosed with PDAC, benign pancreatic cysts, chronic pancreatitis (CP), acute pancreatitis (AP), and other systemic cancers. Healthy controls with no benign or malignant disease were also recruited. PDAC and other cancers were diagnosed by cytological examination of endoscopic ultrasound (EUS)-guided fine-needle aspiration samples or surgical specimens. Pancreatic cysts, CP, and AP were diagnosed by evaluating clinical symptoms and imaging studies (computed tomography, EUS, and magnetic resonance imaging). Patients with evidence of serious illnesses, immunosuppression, or autoimmune/infectious diseases, and those taking immunosuppressive drugs, were excluded.

### 2.3. Sample Collection

Initial blood samples were collected from all subjects at the time of diagnosis. For patients with resectable PDAC, blood samples were collected every month after surgery for 6 months and upon relapse during follow-up. For those with unresectable PDAC, blood samples were collected every 3 months during chemotherapy or radiotherapy. Blood samples were collected only once, at the time of diagnosis, from patients with pancreatic cysts, CP, AP, and other cancers. Peripheral blood samples were also obtained once, at the time of enrollment, from healthy volunteer subjects. Approximately 10 mL blood was collected in ethylenediaminetetraacetic acid tubes and transferred to the laboratory. In addition to the patients’ clinical data, laboratory findings and data on tumor markers, including CA19-9, were recorded. Clinical PDAC staging was performed according to the TNM classification system [10].

### 2.4. Isolation of Total RNA from PBMCs

PBMCs were isolated from whole blood by Ficoll^®^ Paque Plus™ density-gradient centrifugation. Total RNA was extracted from the PBMCs using QIAzol and reverse transcribed to cDNA using PrimeScript™ RT Master Mix according to the manufacturer’s instructions.

### 2.5. Quantitative Real Time PCR of IL-7R Gene Expression

We reported previously that 364 genes are differentially expressed in the PBMCs of patients with PDAC compared with healthy controls (fold change ≥ 2.0 or fold change ≤ 2.0, *p* < 0.05) [9]. We selected 41 of these 364 genes for quantitative real-time polymerase chain reaction (qRT-PCR) verification based on their mononuclear cell specificity and novelty. IL-7R was the factor expressed most consistently in PBMCs from patients with PDAC. qRT-PCR was performed using a PCR detection system (QuantStudio™ Real-Time PCR) and a detection kit (Taqman™ Gene Expression Master Mix). Multiplex qRT-PCR was performed for IL-7R(FAM) and GAPDH(JOE). Quenching was accomplished for all probes using ZEN/IBFQ. The qRT-PCR data were analyzed using the comparative Ct method and normalized to GAPDH. We converted the raw data to relative expression values using to the equations below:IL-7R relative expression value = 2^−ΔCt^ × 1000,
ΔCt = IL-7R Ct − GAPDH Ct.

The primer and probe sequences are listed in Appendix A.

### 2.6. Statistical Analysis

The chi-squared or Fisher’s exact test was used to compare categorical variables. Independent two-sample t tests and analysis of variance were used to compare continuous variables between groups. Post hoc analyses were conducted using the Bonferroni method. To assess associations between two continuous variables, Pearson’s correlation coefficients were calculated. To assess the performance of IL-7R and CA19-9 individually and in combination, receiver operator characteristic curves were constructed and areas under the curve (AUCs) were calculated. The bootstrap method was used to compare AUCs. Diagnostic performance was evaluated on the basis of sensitivity, specificity, accuracy, positive predictive values (PPVs), and negative predictive values (NPVs). A generalized estimation equation was used to compare diagnostic performance. Optimal cutoff values for IL-7R, CA19-9, and IL-7R plus CA19-9 were determined by calculating Youden’s index. Statistical analysis was performed using SAS (version 9.4; SAS Institute, Cary, NC, USA) and R (version 4.0.3; http://www.R-project.org, accessed on 2 January 2020). The significance level was set at *p* < 0.05.

## 3. Results

### 3.1. IL-7R Levels and Subject Characteristics

Of the 529 patients screened, 7 were excluded (5 withdrawn, 1 methotrexate user, 1 PBMC isolation failure), resulting in a final cohort of 522 patients (Figure 1). Initial blood samples were obtained at the time of diagnosis from patients with PDAC (n = 135), healthy controls (n = 100), and patients with pancreatic cysts (n = 82), CP (n = 42), AP (n = 47), and other malignancies 9n = 116; bile duct cancer (n = 58), stomach cancer (n = 11), colon cancer (n = 18), hepatocellular cancer (n = 9), and lung cancer (n = 20)].0

Characteristics did not differ significantly between the PDAC group and the healthy control, pancreatic cyst, CP and AP groups (Table 1), except that the PDAC group was older. The bilirubin, aspartate transaminase, alanine transferase, and CA19-9 levels were higher in the PDAC group than in the other groups, likely due to biliary obstruction by PDAC. The IL-7R mRNA level was significantly higher in the PDAC group than in the healthy control and benign pancreatic disease groups (Figure 2A). Patients with PDAC had significantly higher IL-7R levels than did those with bile duct, stomach, liver, colon, and lung cancers (Figure 2B). CA19-9 is a marker of both pancreatic and biliary tract cancer; we measured CA19-9 levels only in patients with biliary tract cancers. The CA19-9 level did not differ between patients with pancreatic and biliary tract cancers (1946.2 ± 4513.5 vs, 1391.4 ± 1391.4 U/mL, *p* = 0.41).

### 3.2. Diagnostic Performance of IL7-R and/or CA19-9

The IL-7R level did not differ between patients with resectable (n = 55) and unresectable (n = 80) PDAC (Figure 3A). However, the CA 19-9 level was significantly higher in the unresectable PDAC group than in the resectable PDAC group (Figure 3B). Similarly, although the IL-7R level did not differ according to pancreatic cancer stage (Figure 3C), the CA19-9 level was higher in patients with stage IV PDAC (Figure 3D). With increasing tumor size, the IL-7R level exhibited a decreasing trend (Figure 3E) and the CA19-9 level showed an increasing trend (Figure 3F). However, this trend is not statistically significant.

The ‘combination’ refers to IL-7R and CA19-9 and the result was positive when IL-7R > 300.3 or CA19-9 > 37.0. In the resectable PDAC group, the AUCs for IL-7R and CA19-9 did not differ significantly (0.850 and 0.838, respectively), but the AUC for IL-7R plus CA19-9 was significantly greater (0.941; Figure 4A, Table 2). In the unresectable PDAC group, the AUCs for CA19-9 alone and in combination with IL-7R were higher than that for IL-7R (Figure 4B). In the overall (resectable and unresectable) PDAC group, the AUC for IL-7R plus CA19-9 was greater than that for IL-7R or CA19-9 alone (Figure 4C).

The diagnostic performance of IL-7R plus CA19-9 was superior to that of either marker alone (Table 2). The sensitivity, specificity, PPV, NPV, and accuracy for IL-7R plus CA19-9 was 85.9%, 96%, 96.7%, 83.5%, and 90.2%, respectively. Notably, these values were significantly higher than those for IL-7R (58.5%, 92%, 90.8%, 62.2%, and 72.8%, respectively) or CA19-9 (73.3%, 96%, 96.1%, 72.7%, and 83%, respectively) alone (*p* < 0.001). For resectable PDAC, the sensitivity and specificity did not differ significantly between IL-7R and CA19-9. For unresectable PDAC, the sensitivity and specificity of CA19-9 were superior to those of IL-7R.

The IL7-R level decreased over 6 months after surgery or chemotherapy (Figure 5A). CA19-9 also exhibited a decreasing trend after surgery (Figure 5B). In patients with unresectable PDAC, the IL-7R level decreased after chemotherapy (Figure 5C), but the CA19-9 level was unchanged (Figure 5D).

## 4. Discussion

The ability to identify cancers by blood sampling is critical, especially for early-stage, intractable cancers such as PDAC [11]. The goal of early cancer detection is to identify the cancer at a stage when treatment is more likely to be successful, thereby offering better long-term survival. Several research groups have proposed the detection of cancer by blood tests alone or in combination with imaging studies [11,12,13,14].

In this study, the mean IL-7R level was significantly higher in patients with PDAC than in healthy controls, patients with benign pancreatic diseases, and patients with other systemic malignancies. The dense stromal tissue in PDAC tumors contains various inflammatory cell types [4,15] that produce and secrete immunosuppressive cytokines. These cytokines include IL-6, IL-10, IL-13, vascular endothelial growth factor (VEGF), and transforming growth factor-beta (TGF-β), which are collectively presumed to create a favorable environment for PDCA development and progression [16,17]. IL-7R is a heteromeric complex that interacts with IL-7 and comprises the cytokine-receptor gamma chain (IL-2RG and CD132) and IL-7 receptor α chain (IL-7R, IL-7RA, and CD127). IL7R is expressed primarily by T cells (especially memory T cells) and to a lesser extent by B cells, natural killer cells, and monocytes. IL-7R plays an important role in immune responses, including the regulation of cell development and differentiation and maintenance of homeostatic expansion in T-cell subpopulations [18]. IL-7R is also critical for the suppression of T-cell exhaustion [19,20].

We identified 361 significant PDAC markers from mRNA-Seq data from our pilot study [9]. Among them, IL-7R was selected for further study because it is consistently elevated in patients with PDAC compared with healthy individuals. [9] As we focused on a blood PDAC biomarker, the biological mechanism of IL-7R mRNA upregulation in PBMCs from patients with PDAC fell outside the scope of this study. Briefly, IL-7R is critical for lymphocyte development during fetal development and memory T-cell generation in pathological states [21,22]. However, IL-7R mediates inflammatory diseases, including ulcerative colitis [23], rheumatoid arthritis [24], and autoimmune diabetes [25]. Furthermore, the IL-7–IL-7R axis is involved in hematological cancers, including T-cell acute lymphoblastic leukemia [26], chronic lymphocytic leukemia [21], and Hodgkin’s lymphoma [27]. Based on this information, we speculate that the IL-7–IL-7R axis plays roles in tumorigenesis and the maintenance of a healthy immune status.

The feasibility of surgical intervention is a critical factor in PDAC treatment. For this reason, IL-7R expression was compared between patients with operable and non-operable PDAC. Unfortunately, the IL-7R level did not differ significantly between patients with resectable and unresectable tumors. We also investigated the relationship between IL-7R expression and tumor stage. The IL-7R level decreased, and that of CA19-9 increased, with increasing tumor size. Overall, the CA19-9 level increased and the IL-7R level decreased as the cancer progressed. These trends are explained by the role of CA19-9 as a cancer-related marker and IL-7R as a cancer-responsive marker. Because IL-7R represents the host response to cancer, its expression can be host specific and may decrease as the cancer grows.

Relative to IL-7R or CA19-9 alone, the combination of IL-7R and CA19-9 had greater sensitivity (58.5% and 73.3% vs. 85.9%) and a larger AUC (0.766 and 0.881 vs. 0.945). IL-1β, IL-6, IL-8, IL-10, TGF, and VEGF are PDAC-associated cytokines studied in duplicate in more than four studies, and are elevated in pancreatic cancer [8]. Individual cytokines exhibit poor diagnostic performance, whereas cytokine panels have superior diagnostic performance to CA19-9 alone [28]. Furthermore, the combined use of cytokine panels and CA19-9 improves the diagnostic performance [29]. These results suggest that cytokine secretion reflects host immunocompetence during early-stage PDAC. As the cancer progresses, the host may become immunocompromised, leading to decreased cytokine secretion. Conversely, CA19-9 levels increase as PDAC progress, suggesting that immune markers and CA19-9 have a complementary relationship. Thus, a biomarker panel that encompasses immune markers of cancer development and progression may increase the PDAC diagnostic performance.

The IL-7R and CA19-9 levels decreased postoperatively in patients with resectable PDAC. The decreased IL-7R level persisted for 6 months postoperatively. In patients with unresectable PDAC, IL 7R expression was low for 6 months after chemotherapy. The CA19-9 level did not exhibit such a trend. Although the decreased IL-7R level after tumor removal or chemotherapy can be attributed to reduced tumor activity, reduced host immunity as a result of surgery or chemotherapy cannot be ruled out and warrants further research.

In previous works, cytokine levels in patients with PDAC were compared with those in healthy controls or patients with benign pancreatic diseases. In this study, IL-7R expression was lower in patients with systemic cancers than in those with PDAC, suggesting that this expression can be used to differentiate PDAC from other carcinomas. In previous studies, cytokine and inflammatory markers were measured in serum, tissue, plasma, pancreatic fluid, and whole blood [8]. In contrast, we isolated PMBCs to measure their IL-7R expression. The two previous studies involving PBMC analysis differed from the present study [3,30]. We performed cell grouping by single-cell analysis to identify target cells. In addition, we investigated diagnostic performance, which was not performed in the previous works, and concluded this performance is improved by the combined use of CA19-9 and IL-7R as biomarkers.

A limitation of this study is that the contribution of IL-7R to PDAC was not evaluated, despite the identification of this protein as a potential biomarker. In addition, the low sensitivity of IL-7R alone suggests that has only moderate utility as a screening tool. The reduction of the host immune response with PDAC progression is reflected by decreased IL-7R expression. For this reason, the identification of immune markers that are expressed during advanced-stage PDAC is needed. Alternatively, an immune marker that responds to cancer development could be used as an early screening marker. As data on the link between the IL-7R level and PDAC prognosis are insufficient, the potential of this biomarker to predict PDAC recurrence is unclear. Indeed, the accurate measurement of cancer-related immune markers in patients who are immunocompromised as a result of postoperative adjuvant chemotherapy or palliative chemotherapy is difficult. A well-designed prospective randomized study is needed to identify novel immune markers.

## 5. Conclusions

The immune system positively and negatively regulates tumor development and progression, and crosstalk between cancer and immune cells is a hallmark of cancer. The combined use of CA19-9 and IL-7R as biomarkers improved PDAC diagnostic accuracy because the former is secreted by PDACs and the latter is implicated in the immune response to PDAC. Thus, this combination is useful for PDAC detection and the monitoring of its progression. A single cytokine is insufficient as a diagnostic, predictive, or prognostic PDAC biomarker. Instead, a panel of cytokines could be used to distinguish patients with PDAC from healthy individuals, patients with other non-malignant pancreatic diseases, and patients with other systemic malignancies. Further prospective studies are needed to validate and build upon the findings presented here.

## Figures and Tables

**Figure 1 cancers-14-00853-f001:**
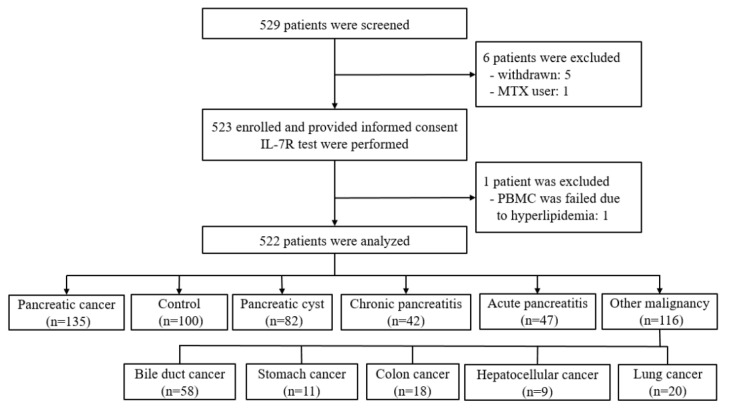
Flow chart of patient selection. MTX, methotrexate; IL-7R, interleukin-7 receptor; PBMC, peripheral blood mononuclear.

**Figure 2 cancers-14-00853-f002:**
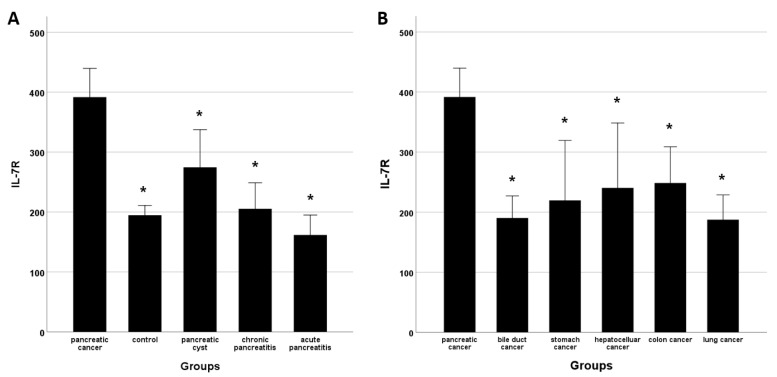
IL-7R levels. (**A**) IL-7R levels were significantly higher in patients with PDAC than in healthy controls and patients with pancreatic cysts, chronic pancreatitis, and acute pancreatitis. (**B**) Patients with PDAC had higher IL-7R levels than did those with systemic malignancies of the bile duct, stomach, liver, colon, or lung. * *p* < 0.005 vs. pancreatic cancer.

**Figure 3 cancers-14-00853-f003:**
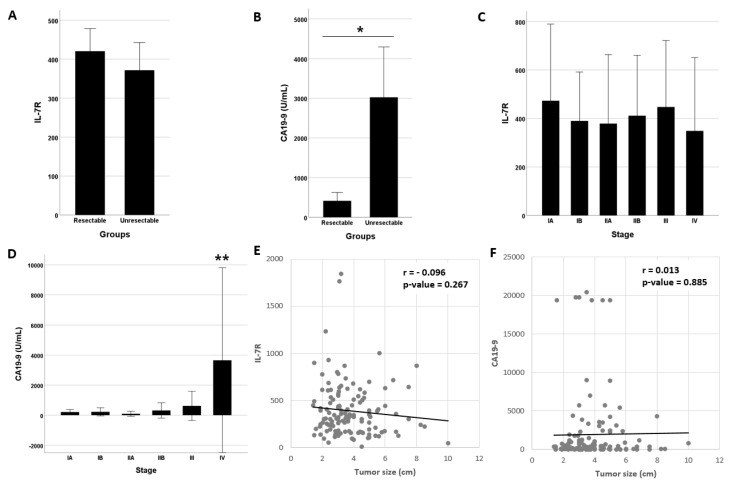
IL-7R and CA19-9 levels in patients with PDAC. (**A**) IL-7R levels did not differ between patients with resectable and unresectable PDAC. (**B**) CA19-9 levels were higher in patients with unresectable PDAC than in those with resectable PDAC. (**C**) IL-7R levels did not differ according to the PDAC stage. (**D**) CA19-9 levels were higher in patients with stage IV PDAC than in those with PDAC of other stages. (**E**) IL-7R levels decreased non-significantly with increasing tumor size. (**F**) CA19-9 levels increased non-significantly with increasing tumor size. * *p* < 0.005, resectable vs. unresectable PDAC. ** *p* < 0.005, stage IV vs. stages I–III PDAC.

**Figure 4 cancers-14-00853-f004:**
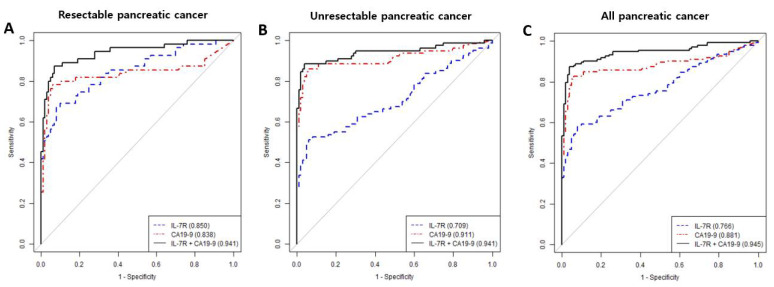
ROC curves for IL-7R, CA19-9, and IL-7R plus CA19-9 for PDAC assessment. (**A**) In the resectable PDAC group, AUCs for IL-7R and CA19-9 did not differ, but the AUC for IL-7R plus CA19-9 was higher than that for IL-7R or CA19-9 alone. (**B**) In the unresectable PDAC group, the AUCs for CA19-9 alone and in combination with IL-7R were higher than that for IL-7R alone. (**C**) In the resectable and unresectable PDAC groups, the AUC for IL-7R combined with CA19-9 was higher than those for IL-7R and CA19-9 alone.

**Figure 5 cancers-14-00853-f005:**
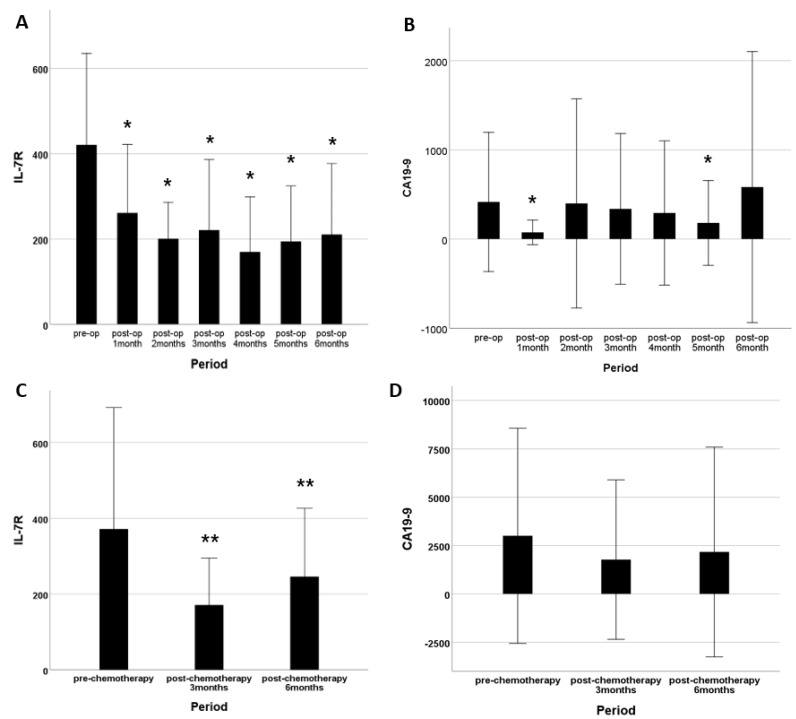
IL-7R and CA19-9 levels after PDAC treatment. (**A**) In the resectable PDAC group, the IL-7R level was lower than preoperatively at 6 months postoperatively. (**B**) In the resectable PDAC group, the CA19-9 level decreased immediately after surgery and fluctuated thereafter. (**C**) In the unresectable PDAC group, the IL-7R level was lower after chemotherapy relative to the pretreatment level. (**D**) In the unresectable PDAC group, the CA19-9 levels measured before and after chemotherapy did not differ. * *p* < 0.005 vs. preoperatively. ** *p* < 0.005 vs. before chemotherapy.

**Table 1 cancers-14-00853-t001:** Characteristics of the study population.

	Pancreatic Cancer (n = 135)	Control (n = 100)	Pancreatic Cyst (n = 82)	Chronic Pancreatitis (n = 42)	Acute Pancreatitis (n = 47)
**Age**, years (mean ± SD)	67.1 ± 11.5	51.9 ± 12.3 ^†^	64.5 ± 10.1	56.7 ± 13.2 ^†^	47.7 ± 15.2 ^†^
**male: female**	70:65	43:50	29:53	32:10	35:12
**BMI**, kg/m^2^ (mean ± SD)	23.0 ± 3.1	24.5 ± 4.1 ^†^	23.3 ± 3.1	23.2 ± 2.8	26.2 ± 4.8 ^†^
**DM**, n (%)	50 (37.0)	NA	19 (23.2)	22 (52.4)	10 (21.3)
**WBC**, count/uL (mean ± SD)	7358.4 ± 3704.7	6041.7 ± 1688.7	5997.4 ± 1837.3	9823.2 ± 14,658.8	8903.0 ± 4680.3
**neutrophil**	4873.1 ± 3290.6	3268.7 ± 1196.2 ^†^	3366.0 ± 1255.6 ^†^	4658.0 ± 2182.9	6417.0 ± 4696.0 ^†^
**lymphocyte**	1669.9 ± 640.1	2135.3 ± 633.0 ^†^	2259.1 ± 666.8 ^†^	2043.2 ± 796.8 ^†^	1590.2 ± 804.1
**monocyte**	605.3 ± 453.1	439.4 ± 166.3 ^†^	461.9 ± 165.0 ^†^	606.6 ± 230.6	683.6 ± 380.3
**eosinophil**	159.8 ± 138.2	154.1 ± 131.2	164.2 ± 194.5	218.0 ± 157.3	191.5 ± 176.6
**basophil**	39.2 ± 22.0	42.7 ± 33.0	39.0 ± 20.3	44.6 ± 22.9	35.7 ± 20.1
**platelet** count 10^3^/μL (mean ± SD)	237.0 ± 82.7	NA	230.2 ± 55.7	248.3 ± 76.3	226.1 ± 78.1
**protein**, g/dL (mean ± SD)	6.8 ± 0.8	7.1 ± 0.4 ^†^	7.0 ± 0.5 ^†^	6.9 ± 0.6	6.2 ± 1.1 ^†^
**albumin**, g/dL (mean ± SD)	3.9 ± 0.4	4.3 ± 0.5 ^†^	4.3 ± 0.3 ^†^	4.1 ± 0.4	3.7 ± 0.6
**bilirubin**, mg/dL (mean ± SD)	2.9 ± 4.6	0.7 ± 0.3 ^†^	0.7 ± 0.3 ^†^	0.7 ± 0.4 ^†^	1.2 ± 0.9 ^†^
**AST**, IU/L (mean ± SD)	91.8 ± 153.0	25.5 ± 10.4 ^†^	27.6 ± 13.3 ^†^	25.2 ± 10.5 ^†^	52.8 ± 68.9
**ALT**, IU/L (mean ± SD)	103.2 ± 173.6	23.2 ± 14.2 ^†^	24.1 ± 17.8 ^†^	21.8 ± 11.2 ^†^	56.7 ± 73.3
**CRP**, mg/L (mean ± SD)	20.4 ± 47.4	NA	2.8 ± 15.9	17.8 ± 44.6	71.6 ± 88.7 ^†^
**CEA**, ng/L (mean ± SD)	98.1 ± 699.9	1.7 ± 1.0	2.3 ± 1.7	3.8 ± 3.4	1.8 ± 1.4
**CA19-9**, U/mL (mean ± SD)	1946.2 ± 4513.5	13.6 ± 39.2 ^†^	25.3 ± 81.8 ^†^	48.2 ± 88.8 ^†^	13.0 ± 14.8 ^†^
**IL-7R**, (mean ± SD)	391.7 ± 281.9	194.7 ± 82.0 ^†^	274.6 ± 286.0 ^†^	205.3 ± 139.8 ^†^	161.8 ± 112.9 ^†^

SD—standard deviation; BMI—body mass index; DM—diabetes mellitus; NA—non-available; WBC—whole blood cell; AST—aspartate transaminase; ALT—alanine transaminase; CRP—c-reactive protein; CEA—carcino-embryonic antigen; CA19-9—carbohydrate antigen 19-9; IL-7R—interleukin-7 receptor. ^†^, *p*-value < 0.005 comparing with pancreatic cancer.

**Table 2 cancers-14-00853-t002:** Diagnostic performance of IL-7R and CA19-9 alone and combination for pancreatic adenocarcinoma (PDAC).

	Marker	AUC	Sensitivity (%)	Specificity (%)	PPV (%)	NPV (%)	Accuracy (%)
**Resectable pancreatic cancer**	IL-7R	0.850	67.3	92.0	82.2	83.6	83.2
	CA19-9	0.838	65.5	96.0	90.0	83.5	85.2
	IL-7R + CA19-9	0.941	87.3	93.0	87.3	93.0	91.0
	IL-7R vs. CA19-9	>0.999	>0.999	0.735	0.823	>0.999	>0.999
	IL-7R vs. combination ^†^	0.014	0.002	>0.999	0.666	0.004	0.011
	CA19-9 vs. combination	0.027	0.003	0.761	>0.999	0.008	0.173
**Unresectable** **pancreatic cancer**	IL-7R	0.709	51.3	94.0	87.2	70.7	75.0
	CA19-9	0.911	78.8	96.0	94.0	85.0	88.3
	IL-7R + CA19-9	0.941	86.3	96.0	94.5	89.7	91.7
	IL-7R vs. CA19-9	<0.001	<0.001	>0.999	0.657	0.001	0.002
	IL-7R vs. combination	<0.001	<0.001	>0.999	0.299	<0.001	<0.001
	CA19-9 vs. combination	0.507	0.157	>0.999	>0.999	0.166	0.319
**All pancreatic cancer**	IL-7R	0.766	58.5	92.0	90.8	62.2	72.8
	CA19-9	0.881	73.3	96.0	96.1	72.7	83.0
	IL-7R + CA19-9	0.945	85.9	96.0	96.7	83.5	90.2
	IL-7R vs. CA19-9	0.017	0.044	0.735	0.412	0.028	0.021
	IL-7R vs. combination	<0.001	<0.001	0.459	0.132	<0.001	<0.001
	CA19-9 vs. combination	0.008	0.001	>0.999	>0.999	0.001	0.004

AUC—area under curve; PPV—positive predictive value; NPV—negative predictive value. Cut-off points: IL-7R >300.3, CA19-9 > 37.0, combination > 0.4962455. ^†^ Combination means IL-7R plus CA19-9 and positive was when the following conditions were satisfied: IL-7R > 300.3 or CA19-9 > 37.0.

## Data Availability

The data used in this study are available in this article.

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
