# Peer review of "Validation of IL-7R as an Immunological Biomarker for Human Pancreatic Ductal Adenocarcinoma"

_cancers, 2022, doi:10.3390/cancers14030853_

Round 1
Reviewer 1 Report
This manuscript aims to enhance the diagnostic accuracy of ductal pancreatic adenocarcinoma (PDAC) by detecting IL-7R and CA19-9 mRNA levels as biomarkers in peripheral blood mononuclear cells (PBMCs) in patients. The study is well conducted. Some suggestions are as follows:
- Not sure if the manuscript should include "supplementary results" following the text (pages 10 and 11) in this journal?
- Are there methods or biomarkers for the diagnosis of PDAC? Please compare their benefits and disadvantages with the sensing of the combined use of IL-7R and CA19-9.
- Do the authors provide the information, such as Figure 2, on the levels of CA19-9 in healthy controls and patients with pancreatic cancer as well as those with different cancers?
Author Response
â–£ For Reviewer 1
We appreciate your thoughtful comments regarding our manuscript.
â–£ Evaluations
Comments by the reviewers:
Reviewer 1.
This manuscript aims to enhance the diagnostic accuracy of ductal pancreatic adenocarcinoma (PDAC) by detecting IL-7R and CA19-9 mRNA levels as biomarkers in peripheral blood mononuclear cells (PBMCs) in patients. The study is well conducted. Some suggestions are as follows:
- Not sure if the manuscript should include "supplementary results" following the text (pages 10 and 11) in this journal?
Answer) Exploration and basic research on IL-7R has published the results in a previous paper (Heo, S.-H.; Jang, S.I.; Kim, S.Y.; Choi, B.; Lee, D.K.; Lee, H.K.; Chang, E.-J. Characterization of circulating il-7r positive cell populations for early detection of pancreatic ductal adenocarcinoma. Journal of Clinical Medicine 2021, 10, 4157) and we cite a previous work on basic IL-7R research (reference 9) in the paper. This clinical trial was based on that study. The basic research contents underlying this clinical trial can be checked through the published article (reference 9). There are thus no Supplementary Results
- Are there methods or biomarkers for the diagnosis of PDAC? Please compare their benefits and disadvantages with the sensing of the combined use of IL-7R and CA19-9.
Answer) PDAC is diagnosed via EUS-guided fine-needle biopsy or surgical biopsy based on imaging data (CT, MRI, EUS, or PET-CT). We explore whether IL-7R might serve as a diagnostic PDAC biomarker. Indeed, previous studies sought such biomarkers. In the Discussion, we write: “Previous studies measured cytokine and inflammatory markers in serum, tissue, plasma, pancreatic fluid, and whole blood [8]”. However, no commercial kit for biomarker detection in the clinic or research laboratory is yet available. The differences between this study and previous studies are described in the Discussion as follows:
‘In contrast, we isolated PMBCs to measure their IL-7R expression. The two previous studies involving PBMC analysis differed from the present study [3,30]. We performed cell grouping by single-cell analysis to identify target cells. In addition, we investigated diagnostic performance, which was not performed in the previous works, and concluded this performance is improved by the combined use of CA19-9 and IL-7R as biomarkers.’
Reference
3 Bellone, G.; Novarino, A.; Vizio, B.; Brondino, G.; Addeo, A.; Prati, A.; Giacobino, A.; Campra, D.; Fronda, G.R.; Ciuffreda, L. Impact of surgery and chemotherapy on cellular immunity in pancreatic carcinoma patients in view of an integration of standard cancer treatment with immunotherapy. International journal of oncology 2009, 34, 1701-1715.
8 Yako, Y.Y.; Kruger, D.; Smith, M.; Brand, M. Cytokines as biomarkers of pancreatic ductal adenocarcinoma: A systematic review. Plos One 2016, 11.
30 Baine, M.J.; Menning, M.; Smith, L.M.; Mallya, K.; Kaur, S.; Rachagani, S.; Chakraborty, S.; Sasson, A.R.; Brand, R.E.; Batra, S.K. Differential gene expression analysis of peripheral blood mononuclear cells reveals novel test for early detection of pancreatic cancer. Cancer biomarkers : section A of Disease markers 2011, 11, 1-14.
As the reviewer points out, the advantages and disadvantages of the combined use of IL-7R and CA19-9 status to diagnose PDAC are difficult to compare when both imaging and biopsy studies are included. These modalities differ markedly; biopsy affords the final confirmation. We deal with this in the Discussion.
‘In contrast, we isolated PMBCs to measure their IL-7R expression. The two previous studies involving PBMC analysis differed from the present study [3,30]. We performed cell grouping by single-cell analysis to identify target cells. In addition, we investigated diagnostic performance, which was not performed in the previous works, and concluded this performance is improved by the combined use of CA19-9 and IL-7R as biomarkers.’
And it is considered that the comparison with CA19-9, which is currently used clinically, was sufficiently compared as CA19-9 alone and the combined use of IL-7R and CA19-9 were also compared in this paper.
3.Do the authors provide the information, such as Figure 2, on the levels of CA19-9 in healthy controls and patients with pancreatic cancer as well as those with different cancers?
Answer) Table 1 lists the CA 19-9 levels; those of the control, pancreatic cancer, pancreatic cyst, and chronic and acute pancreatitis groups differed.
Table 1. Characteristics of the study population.
|
|
Pancreatic cancer (n=135) |
Control (n=100) |
Pancreatic cyst (n=82) |
Chronic pancreatitis (n=42) |
Acute pancreatitis (n=47) |
|
Age, years (mean ± SD) |
67.1 ± 11.5 |
51.9 ± 12.3† |
64.5 ± 10.1 |
56.7 ± 13.2† |
47.7 ± 15.2† |
|
Male : female |
70:65 |
43:50 |
29:53 |
32:10 |
35:12 |
|
BMI, kg/m2 (mean ± SD) |
23.0 ± 3.1 |
24.5 ± 4.1† |
23.3 ± 3.1 |
23.2 ± 2.8 |
26.2 ± 4.8† |
|
DM, n (%) |
50 (37.0) |
NA |
19 (23.2) |
22 (52.4) |
10 (21.3) |
|
WBC, count/uL (mean ± SD) |
7358.4 ± 3704.7 |
6041.7 ± 1688.7 |
5997.4 ± 1837.3 |
9823.2 ± 14658.8 |
8903.0 ± 4680.3 |
|
Neutrophil |
4873.1 ± 3290.6 |
3268.7 ± 1196.2† |
3366.0 ± 1255.6† |
4658.0 ± 2182.9 |
6417.0 ± 4696.0† |
|
Lymphocyte |
1669.9 ± 640.1 |
2135.3 ± 633.0† |
2259.1 ± 666.8† |
2043.2 ± 796.8† |
1590.2 ± 804.1 |
|
Monocyte |
605.3 ± 453.1 |
439.4 ± 166.3† |
461.9 ± 165.0† |
606.6 ± 230.6 |
683.6 ± 380.3 |
|
Eosinophil |
159.8 ± 138.2 |
154.1 ± 131.2 |
164.2 ± 194.5 |
218.0 ± 157.3 |
191.5 ± 176.6 |
|
Basophil |
39.2 ± 22.0 |
42.7 ± 33.0 |
39.0 ± 20.3 |
44.6 ± 22.9 |
35.7 ± 20.1 |
|
Platelet count 103/ μL (mean ± SD) |
237.0 ± 82.7 |
NA |
230.2 ± 55.7 |
248.3 ± 76.3 |
226.1 ± 78.1 |
|
Protein, g/dL (mean ± SD) |
6.8 ± 0.8 |
7.1 ± 0.4† |
7.0 ± 0.5† |
6.9 ± 0.6 |
6.2 ± 1.1† |
|
Albumin, g/dL (mean ± SD) |
3.9 ± 0.4 |
4.3 ± 0.5† |
4.3 ± 0.3† |
4.1 ± 0.4 |
3.7 ± 0.6 |
|
Bilirubin, mg/dL (mean ± SD) |
2.9 ± 4.6 |
0.7 ± 0.3† |
0.7 ± 0.3† |
0.7 ± 0.4† |
1.2 ± 0.9† |
|
AST, IU/L (mean ± SD) |
91.8 ± 153.0 |
25.5 ± 10.4† |
27.6 ± 13.3† |
25.2 ± 10.5† |
52.8 ± 68.9 |
|
ALT, IU/L (mean ± SD) |
103.2 ± 173.6 |
23.2 ± 14.2† |
24.1 ± 17.8† |
21.8 ± 11.2† |
56.7 ± 73.3 |
|
CRP, mg/L (mean ± SD) |
20.4 ± 47.4 |
NA |
2.8 ± 15.9 |
17.8 ± 44.6 |
71.6 ± 88.7† |
|
CEA, ng/L (mean ± SD) |
98.1 ± 699.9 |
1.7 ± 1.0 |
2.3 ± 1.7 |
3.8 ± 3.4 |
1.8 ± 1.4 |
|
CA19-9, U/mL (mean ± SD) |
1946.2 ± 4513.5 |
13.6 ± 39.2† |
25.3 ± 81.8† |
48.2 ± 88.8† |
13.0 ± 14.8† |
|
IL-7R, (mean ± SD) |
391.7 ± 281.9 |
194.7 ± 82.0† |
274.6 ± 286.0† |
205.3 ± 139.8† |
161.8 ± 112.9† |
. †Combination means IL-7R plus CA19-9 and positive is when the following conditions are satisfied: IL-7R>300.3 or CA19-9>37.0.
CA19-9 serves a marker of pancreatic and biliary tract cancers; we measured CA19-9 levels in (only) two carcinomas. The result is as follows.
|
|
Pancreatic cancer (n=135) |
Bile duct cancer (n=57) |
P-value |
|
CA19-9, U/mL (mean ± SD) |
1946.2 ± 4513.5 |
1391.4 ±1391.4 |
0.410 |
The CA19-9 level did not differ between patients with pancreatic and biliary tract cancer. However, the IL-7R levels did. As the reviewer points out, it is difficult to show the IL-7R levels of all other carcinomas in a Figure. We adopt the reviewer’s suggestion; we now comment on the data (in the Results) as follows:
“CA19-9 is a marker of both pancreatic and biliary tract cancer; we measured CA19-9 levels only in patients with biliary tract cancers. The CA19-9 level did not differ between patients with pancreatic and biliary tract cancers (1,946.2 ± 4,513.5 vs, 1,391.4 ± 1,391.4 U/mL, p = 0.41).”
Reviewer 2 Report
This manuscript suggests that the combined use of CA19-9 and IL-7R as biomarkers improved PDAC diagnostic accuracy. More importantly, it was supposed that a panel of cytokines could be used to distinguish patients with PDAC from healthy individuals, patients with other non-malignant pancreatic diseases, and patients with other systemic malignancies. Overall, the text is clear and well-written, and the analysis has been done carefully. However, to be considered for publication, the paper requires some improvements in the methodological process and discussion.
- It was concluded from figures 3E and 3F that the IL-7R level exhibited a decreasing trend and the CA19-9 level showed an increasing trend with increasing tumor size. And this conclusion supports that decreased IL-7R level as cancer progressed was a host response to cancer. However, there are several problems. Firstly, the trends are too non-significant to certify the conclusions. Secondly, most patients have smaller tumor size rather than relatively evenly distracted, which bring higher contingency. Last but not least, tumor size can not represent the tumor progression, such as the higher CA19-9 level in stage IV PDAC but non-significantly higher in patients with larger tumor size. Therefore, it is not clearly certified that IL-7R decrease can be related to cancer progression.
- According to the manuscript, the combination of IL-7R and CA19-9 could improve the diagnostic performance. However, how to combine these two biomarkers has not been illustrated. Is it enough for either one to rise, or is it more valuable for both to rise?
Author Response
â–£ For Reviewer 2
We appreciate your thoughtful comments regarding our manuscript.
â–£ Evaluations
Comments by the reviewers:
Reviewer 2
This manuscript suggests that the combined use of CA19-9 and IL-7R as biomarkers improved PDAC diagnostic accuracy. More importantly, it was supposed that a panel of cytokines could be used to distinguish patients with PDAC from healthy individuals, patients with other non-malignant pancreatic diseases, and patients with other systemic malignancies. Overall, the text is clear and well-written, and the analysis has been done carefully. However, to be considered for publication, the paper requires some improvements in the methodological process and discussion.
- It was concluded from figures 3E and 3F that the IL-7R level exhibited a decreasing trend and the CA19-9 level showed an increasing trend with increasing tumor size. And this conclusion supports that decreased IL-7R level as cancer progressed was a host response to cancer. However, there are several problems. Firstly, the trends are too non-significant to certify the conclusions. Secondly, most patients have smaller tumor size rather than relatively evenly distracted, which bring higher contingency. Last but not least, tumor size can not represent the tumor progression, such as the higher CA19-9 level in stage IV PDAC but non-significantly higher in patients with larger tumor size. Therefore, it is not clearly certified that IL-7R decrease can be related to cancer progression.
Answer) We agree. Statistical significance is lacking; it is not possible to draw conclusions because only an overall trend is apparent. Even in patient 3C, the IL-7R level did not differ by stage. The Figure shows only that the IL-7R tends to decrease with larger tumor size.
Table 2 compares the data on patients with resectable and unresectable pancreatic cancers. The IL-7R levels were higher in the former patients and the CA 19-9 levels higher in the latter. The combination of the two markers afforded a higher diagnostic performance than did either marker alone. IL-7R is a marker of cancer response; CA19-9 is a marker of cancer cell death.
Table 2. Diagnostic performance of IL-7R and CA19-9 alone and combination for pancreatic adenocarcinoma (PDAC).
|
  |
Marker |
AUC |
Sensitivity(%) |
Specificity(%) |
PPV(%) |
NPV(%) |
Accuracy(%) |
|
Resectable pancreatic cancer
  |
IL-7R |
0.850 |
67.3 |
92.0 |
82.2 |
83.6 |
83.2 |
|
CA19-9 |
0.838 |
65.5 |
96.0 |
90.0 |
83.5 |
85.2 |
|
|
IL-7R+CA19-9 |
0.941 |
87.3 |
93.0 |
87.3 |
93.0 |
91.0 |
|
|
IL-7R vs CA19-9 |
>.999 |
>.999 |
0.735 |
0.823 |
>.999 |
>.999 |
|
|
IL-7R vs Combination† |
0.014 |
0.002 |
>.999 |
0.666 |
0.004 |
0.011 |
|
|
CA19-9 vs Combination |
0.027 |
0.003 |
0.761 |
>.999 |
0.008 |
0.173 |
|
|
Unresectable pancreatic cancer
  |
IL-7R |
0.709 |
51.3 |
94.0 |
87.2 |
70.7 |
75.0 |
|
CA19-9 |
0.911 |
78.8 |
96.0 |
94.0 |
85.0 |
88.3 |
|
|
IL-7R+CA19-9 |
0.941 |
86.3 |
96.0 |
94.5 |
89.7 |
91.7 |
|
|
IL-7R vs CA19-9 |
<.001 |
<.001 |
>.999 |
0.657 |
0.001 |
0.002 |
|
|
IL-7R vs Combination |
<.001 |
<.001 |
>.999 |
0.299 |
<.001 |
<.001 |
|
|
CA19-9 vs Combination |
0.507 |
0.157 |
>.999 |
>.999 |
0.166 |
0.319 |
|
|
All pancreatic cancer
  |
IL-7R |
0.766 |
58.5 |
92.0 |
90.8 |
62.2 |
72.8 |
|
CA19-9 |
0.881 |
73.3 |
96.0 |
96.1 |
72.7 |
83.0 |
|
|
IL-7R+CA19-9 |
0.945 |
85.9 |
96.0 |
96.7 |
83.5 |
90.2 |
|
|
IL-7R vs CA19-9 |
0.017 |
0.044 |
0.735 |
0.412 |
0.028 |
0.021 |
|
|
IL-7R vs Combination |
<.001 |
<.001 |
0.459 |
0.132 |
<.001 |
<.001 |
|
|
CA19-9 vs Combination |
0.008 |
0.001 |
>.999 |
>.999 |
0.001 |
0.004 |
AUC, area under curve; PPV, positive predictive value; NPV, negative predictive value. Cut-off points: IL-7R >300.3, CA19-9>37.0, Combination>0.4962455. †Combination means IL-7R plus CA19-9 and positive is when the following conditions are satisfied: IL-7R>300.3 or CA19-9>37.0.
In previous basic experiments, it was shown earlier that the IL-7R level increased in the early stages of pancreatic cancer (reference9: Heo, S.-H.; Jang, S.I.; Kim, S.Y.; Choi, B.; Lee, D.K.; Lee, H.K.; Chang, E.-J. Characterization of circulating il-7r positive cell populations for early detection of pancreatic ductal adenocarcinoma. Journal of Clinical Medicine 2021, 10, 4157.) (Figure 2).
Figure 2. Elevation of IL-7Ra expressing cells during early tumorigenesis in vivo. (A) Schematic illustration of an orthotopic syngeneic pancreatic cancer murine model. Pan02 cells were transplanted into pancreas. Four mouse bearing pancreatic cancer per group were sacrificed after 2, 4, 5, 7, 11 days. Mice injected with PBS was used as a naïve control (N = 15). (B,C) On the day of sacrifice, tumors (B) and spleens (C) were removed and weighted in syngeneic mouse model. *** p < 0.001 versus day 2 (B) or naïve control (C). (D) The number of IL-7R expressing cells were determined from mouse PBMC during tumorigenesis in vivo. * p < 0.05, ** p < 0.005, or *** p < 0.001 versus naïve control. (E) IL-7R expression in PBMC was analyzed from naïve or tumor bearing mice after 11 days of Pan02 injection.
Clinical trial revealed that IL-7R expression was high in patients with resectable pancreatic cancer but decreased as cancer advanced. A limitation of our work is that the experiments were performed (only) 11 days after pancreatic cancer development. We thus lack data on changes in IL-7R levels as cancer progresses; we cannot confirm the clinical results. Further work is needed. In summary, we interpret the IL-7R expression level by reference to the results of clinical trials. We do not focus on cancer size alone. The work constitutes a clinical analogy.
Therefore, as your recommendations, we add following contents to the results section
“However, this trend is not statistically significant.”
- According to the manuscript, the combination of IL-7R and CA19-9 could improve the diagnostic performance. However, how to combine these two biomarkers has not been illustrated. Is it enough for either one to rise, or is it more valuable for both to rise?
Answer) The ROC curve identified an IL-7R cut-off of 303.3; a higher level indicated positivity. For CA19-9, a figure higher than the commercial cut-off (37) was considered positive. If even one of the two markers was positive, the combination was taken to be positive, as shown in Table 2. This is expressed in Table 2 as follows.
‘†Combination means IL-7R plus CA19-9 and positive is when the following conditions are satisfied: IL-7R>300.3 or CA19-9>37.0.’
Table 2. Diagnostic performance of IL-7R and CA19-9 alone and combination for pancreatic adenocarcinoma (PDAC).
|
  |
Marker |
AUC |
Sensitivity(%) |
Specificity(%) |
PPV(%) |
NPV(%) |
Accuracy(%) |
|
Resectable pancreatic cancer
  |
IL-7R |
0.850 |
67.3 |
92.0 |
82.2 |
83.6 |
83.2 |
|
CA19-9 |
0.838 |
65.5 |
96.0 |
90.0 |
83.5 |
85.2 |
|
|
IL-7R+CA19-9 |
0.941 |
87.3 |
93.0 |
87.3 |
93.0 |
91.0 |
|
|
IL-7R vs CA19-9 |
>.999 |
>.999 |
0.735 |
0.823 |
>.999 |
>.999 |
|
|
IL-7R vs Combination† |
0.014 |
0.002 |
>.999 |
0.666 |
0.004 |
0.011 |
|
|
CA19-9 vs Combination |
0.027 |
0.003 |
0.761 |
>.999 |
0.008 |
0.173 |
|
|
Unresectable pancreatic cancer
  |
IL-7R |
0.709 |
51.3 |
94.0 |
87.2 |
70.7 |
75.0 |
|
CA19-9 |
0.911 |
78.8 |
96.0 |
94.0 |
85.0 |
88.3 |
|
|
IL-7R+CA19-9 |
0.941 |
86.3 |
96.0 |
94.5 |
89.7 |
91.7 |
|
|
IL-7R vs CA19-9 |
<.001 |
<.001 |
>.999 |
0.657 |
0.001 |
0.002 |
|
|
IL-7R vs Combination |
<.001 |
<.001 |
>.999 |
0.299 |
<.001 |
<.001 |
|
|
CA19-9 vs Combination |
0.507 |
0.157 |
>.999 |
>.999 |
0.166 |
0.319 |
|
|
All pancreatic cancer
  |
IL-7R |
0.766 |
58.5 |
92.0 |
90.8 |
62.2 |
72.8 |
|
CA19-9 |
0.881 |
73.3 |
96.0 |
96.1 |
72.7 |
83.0 |
|
|
IL-7R+CA19-9 |
0.945 |
85.9 |
96.0 |
96.7 |
83.5 |
90.2 |
|
|
IL-7R vs CA19-9 |
0.017 |
0.044 |
0.735 |
0.412 |
0.028 |
0.021 |
|
|
IL-7R vs Combination |
<.001 |
<.001 |
0.459 |
0.132 |
<.001 |
<.001 |
|
|
CA19-9 vs Combination |
0.008 |
0.001 |
>.999 |
>.999 |
0.001 |
0.004 |
- AUC, area under curve; PPV, positive predictive value; NPV, negative predictive value. Cut-off points: IL-7R >300.3, CA19-9>37.0, Combination>0.4962455. †Combination means IL-7R plus CA19-9 and positive is when the following conditions are satisfied: IL-7R>300.3 or CA19-9>37.0.
As the reviewer suggested, we have revised the Methods:
In method
“The ‘combination’ refers to IL-7R and CA19-9 and the result was positive when IL-7R > 300.3 or CA19-9 > 37.0”
Reviewer 3 Report
In this manuscript, the author assessed the mRNA levels of interleukin-7R (IL-7R) in 135 PDAC patients and other related control groups and found that the IL-7R level was significantly higher in patients with PDAC than other patients. They further analyzed the diagnostic parameters and showed that the combined use of IL-41 7R and CA19-9 enhanced the diagnostic performance. The results are interesting and clinic-relevant, but they are not solid and lack novelty. Overall, the conclusion is slightly supported by the data and the manuscript is considered to be not suitable for Cancers.
Main comments
- In Figure 1, the differences among PDAC and other groups are quite moderate if using the control value as 1. What is the rationale to amplify the value by 1000?
- Did the author detect the protein levels of IL-7R and CA19-9 in plasma or serum from the sample? It will be interesting and valuable to know the data.
- In line 289, the author stated, “We performed cell grouping by single-cell analysis to identify target 289 cells.” But the reviewer doesn’t see any single-cell data in the manuscript. Actually, the Isoplexis company has developed this type of essay on chip-based cytokines ELISA called single-cell proteomics.
Author Response
â–£ For Reviewer 3
We appreciate your thoughtful comments regarding our manuscript.
â–£ Evaluations
Comments by the reviewers:
Reviewer 3
In this manuscript, the author assessed the mRNA levels of interleukin-7R (IL-7R) in 135 PDAC patients and other related control groups and found that the IL-7R level was significantly higher in patients with PDAC than other patients. They further analyzed the diagnostic parameters and showed that the combined use of IL-41 7R and CA19-9 enhanced the diagnostic performance. The results are interesting and clinic-relevant, but they are not solid and lack novelty. Overall, the conclusion is slightly supported by the data and the manuscript is considered to be not suitable for Cancers.
Main comments
- In Figure 1, the differences among PDAC and other groups are quite moderate if using the control value as 1. What is the rationale to amplify the value by 1000?
Answer) Figure 1 is the Patients flowchart. What the reviewer pointed out seems to be Figure 2. The level in Figure 2 are measured through the formula below.
IL-7R relative expression value = 2-ΔCt × 1000,
ΔCt = IL-7R Ct - GAPDH Ct.
As the reviewer points out, the [IL-7R Ct minus the GAPDH Ct.] was multiplied by 1,000 to convert it to an integer. A value expressed in decimal places is multiplied by 1000 to convert it to an integer value in order to accurately represent values expressed in decimal places. This method converts decimals to integers in a similar way that applies to CA19-9. This was applied equally to IL-7R measured in all cases enrolled in this study. In the statistical data before and after multiplying by 1000, the p-values were not affected by the 1,000-fold multiplication.
- Did the author detect the protein levels of IL-7R and CA19-9 in plasma or serum from the sample? It will be interesting and valuable to know the data.
Answer) Serum CA19-9 levels were measured using a commercial kit. The levels of mRNA encoding IL-7R were measured in PBMC IL-7R-expressing cells isolated from PBMC via qRT-PCR, not in serum or plasma. This part is described in the method as follows.
‘IL-7R was the factor expressed most consistently in PBMCs from patients with PDAC. qRT-PCR was performed using a PCR detection system (QuantStudio™ Real-Time PCR) and a detection kit (Taqman™ Gene Expression Master Mix). Multiplex qRT-PCR was performed for IL-7R(FAM) and GAPDH(JOE). Quenching was accomplished for all probes using ZEN/IBFQ. The qRT-PCR data were analyzed using the comparative Ct method and normalized to GAPDH. We converted the raw data to relative expression values using to the equations below:
IL-7R relative expression value = 2-ΔCt × 1000,
ΔCt = IL-7R Ct - GAPDH Ct.
The primer and probe sequences are listed in Supplementary Table S1.’
- In line 289, the author stated, “We performed cell grouping by single-cell analysis to identify target 289 cells.” But the reviewer doesn’t see any single-cell data in the manuscript. Actually, the Isoplexis company has developed this type of essay on chip-based cytokines ELISA called single-cell proteomics.
Answer) The single cell analyses were reported in previous basic experiments (reference9: Heo, S.-H.; Jang, S.I.; Kim, S.Y.; Choi, B.; Lee, D.K.; Lee, H.K.; Chang, E.-J. Characterization of circulating il-7r positive cell populations for early detection of pancreatic ductal adenocarcinoma. Journal of Clinical Medicine 2021, 10, 4157.) (Figure 1). Reference on the data of basic research is cited in this manuscript (reference 9).
Figure 1. Transcriptomic analysis of PBMCs from normal individuals and pancreatic cancer patients for biomarker discovery. (A) Heatmap showing relative gene expression [log2FPKM (fragments per kilobase million)] of 364 differentially expressed genes (DEGs) in PBMC between 7 normal healthy donors and 15 pancreatic cancer patients. Yellow and blue reflect high and low expression levels, respectively, as indicated in the scale bar (log 2 transformed scale). Each row represents a DEG, and each column represents a sample. List of differentially expressed genes are presented. (B) Multidimensional scaling of transcriptome analyses in these cells. Principal component analysis of transcriptome of healthy individuals
(blue) and pancreatic cancer human patients (red), using gene expression counts for each group. Each dot represents one sample. (C) Top 20 biological processes associated with genes differentially expressed in PBMCs from pancreatic cancer patients. *** p < 0.001 versus healthy controls. (D) Elevated mRNA expression of IL-7R in PBMCs from pancreatic cancer (PC) patients as revealed by transcriptomic analysis. * p < 0.05 versus control. (E) Increased IL-7R positive cells in PBMCs from pancreatic cancer (PC) patients compared with healthy controls (Control). ** p < 0.01 versus control.
Brief details are written in the method as follows;
‘We used omics and bioinformatics techniques to identify blood factors with clinical implications for PDAC detection and interleukin-7R (IL-7R) was discovered as a PDAC-specific immunologic marker by total mRNA sequencing (mRNA-Seq) by a previous study [9].
Reference
- Heo, S.-H.; Jang, S.I.; Kim, S.Y.; Choi, B.; Lee, D.K.; Lee, H.K.; Chang, E.-J. Characterization of circulating il-7r positive cell populations for early detection of pancreatic ductal adenocarcinoma. Journal of Clinical Medicine 2021, 10, 4157.’
Round 2
Reviewer 3 Report
I am interested in the protein level of IL-7R like CA19-9 the author did.
Author Response
â–£ For Reviewer 3
We appreciate your thoughtful comments regarding our manuscript.
â–£ Evaluations
Comments by the reviewers:
Reviewer 3
I am interested in the protein level of IL-7R like CA19-9 the author did.
Answer) According to the reviewer's opinion, it is thought to be important to measure the protein level of IL-7R. In this study, the levels of mRNA encoding IL-7R were measured in PBMC IL-7R-expressing cells isolated from PBMC via qRT-PCR. It is essential to measure the protein synthesized by these mRNAs. Although only the mRNA level was measured in the previous basic study and this study, we will calculate the protein level of IL-7R through additional studies as the reviewer's recommendation.
